# Effects of grazing on the allocation of mass of soil aggregates and aggregate-associated organic carbon in an alpine meadow

Jiwei Wang, Chengzhang Zhao*, Lianchun Zhao, Jun Wen, Qun Li

College of Geography and Environmental Science, Northwest Normal University, Lanzhou, China

* zhaocz601@163.com

**Data Availability Statement:** All relevant data are within the paper and its Supporting Information files.

## Abstract

Soil aggregation is closely related to the soil organic carbon sequestration, both of which plays an important role in the stability of the soil carbon pool. However, the results of the impact of yak grazing intensity on the soil carbon process in alpine meadows have been unclear. With the marsh meadow as the research object in the Gahai wetland of the east Qinghai-Tibet Plateau, we analyzed the influence of different grazing intensities on the allocation of mass, stability and aggregate-associated organic carbon content of aggregates in the surface soil (0-20cm) of pasture by the Le Bissonnais method. The results showed that the mass of aggregates in the particle size class of > 0.2-mm was the highest among the pastures with different grazing intensities. Compared with the no grazing grassland, light grazing promoted the formation of macro aggregates in the particle size class of > 1-mm and improved the stability of soil aggregates. The degree of soil agglomerations and stability of aggregates decreased, and the macro aggregates gradually transformed into micro aggregates (< 0.05-mm class) in moderately and heavily grazed pastures. The > 2-mm and < 0.05-mm classes of particle size had a strong fixation effect on organic carbon. Light grazing promoted the accumulation of organic carbon in this particle size aggregate, and moderate and heavy grazing accelerated the decomposition of organic carbon. There was no significant difference in organic carbon in other particle size aggregates among different grasslands (P > 0.05). This result shows that light grazing, which is a reasonable yak grazing intensity in the study area, is conducive to the formation of a good soil structure in the area and improves the soil carbon sequestration capacity.

## Introduction

The Qinghai-Tibet Plateau is an ideal area to study the feedback effect of carbon cycling and environmental changes in high-altitude ecosystems [1–4]. The swamp meadows are an important part of the alpine meadows on the Qinghai-Tibet Plateau. Their ecosystem is unique; they possess the structure and function of both swamp and grassland ecosystems. Because of their high productivity and low decomposition rate, swamp meadows have the highest soil organic carbon content in the world [1, 5]. However, this type of alpine ecosystem is very vulnerable to

**Funding:** The author(s) received no specific funding for this work.

**Competing interests:** The authors have declared that no competing interests exist.

climate change and human activities. The dynamic changes in the soil carbon pool of alpine ecosystems are considered a typical representative of terrestrial ecosystems responding to the global carbon cycle [6–8]. Over the past few decades, the alpine grassland on the Qinghai-Tibet Plateau has experienced severe degradation, but the driving factors of grassland ecosystem degradation and their impacts have not been determined [9–11]. Some researchers believe that grazing may be the main factor that has led to changes in the productivity, soil structure and soil organic carbon in these alpine grasslands relative to the impact of climate change [12]. However, research findings on the effects of grazing on the soil structure and soil carbon pool have not been uniform, and the results are quite different [13].

Soil aggregates are an important part of soil. The organic matter contained in aggregates can improve the quality and productivity of soil, which is highly important for maintaining the stability of soil structure and the retention of soil nutrients [14–15]. Soil organic carbon is one of the most important factors that affect soil structure and stability and the soil aggregation can cause spatial separation of organisms and organic carbon, which is considered to be an important physical protection mechanism of organic carbon sequestration [16]. Therefore, the stability of soil aggregates determines to some extent the stability of soil organic carbon. In the context of global climate change, more and more attention has been paid to the formation, stability mechanism and main influencing factors of soil aggregates [17]. Many studies have shown that the size and allocation of mass of soil aggregates affect the content and stability of organic carbon. The soil organic carbon stocks is closely related to the stability of aggregates [13]. Currently, most scholars divide soil aggregates into macro aggregates (particle sizes > 0.25-mm) and micro aggregates (particle sizes < 0.25-mm) (usually macro aggregates and micro aggregates are subdivided according to the actual situation), and aggregates with different particle sizes have different abilities in stabilizing soil structure and protecting organic carbon [18]. Many scholars like Elliott [19] believed that the organic carbon associated with macro aggregates is easier to mineralize than that in micro aggregates. The organic carbon in macro aggregates participates in the carbon cycle between vegetation, soil and atmosphere, which is one of the mechanisms of soil organic carbon loss. Scholars like Puget [20] believe that micro aggregates form macro aggregates through the cementing of organic carbon and that land use patterns affect the transformation and redistribution between micro aggregates and macro aggregates.

In recent years, many studies have found that grazing has an important effect on soil aggregates and organic carbon in grassland ecosystems [21]. Livestock causes changes in vegetation, soil physical and chemical properties and microflora through the selective feeding of plants, trampling of grasslands, and return of excreta to grasslands, which affects the formation process of soil aggregates, the ratio of input and output of carbon and existence form or stabilization level of organic carbon in soil [22]. Due to the differences in grassland types, soil properties, grazing history (e.g., grazing intensity, frequency, duration, livestock type, etc.) and grassland management measures, the related research results vary greatly. For example, most studies have shown that grazing can cause soil aggregate to break down, leading to the decrease in the distribution ratio of macro aggregates and the stability of aggregates, which affects the organic carbon sequestration to aggregates [23]. Other studies have also shown that the soil organic carbon content increases first and then decreases gradually[24], or they have showen a decreasing trend [25] or a gradual increase [26] with the increase in grazing intensity in typical grassland areas; Some grazing experiments also confirmed that grazing has no significant effect on the content of organic carbon in soil aggregates with different particle sizes [27]. Milchunas and others [28] also compared the data of enclosure and grazing grasslands in 236 research sites worldwide and they found that there is a complex correlation among grazing, soil properties and soil organic carbon, with both positive and negative correlations. The results of the above

studies indicate that the responses of soil aggregates and soil organic carbon to grazing differ greatly in different grassland ecosystems, even within the same grassland ecosystem. There are many studies on the influence of grazing disturbance on alpine meadows in the Qinghai-Tibet Plateau, which mostly involve the changes of characteristic in plant community, soil physical and chemical properties and the soil carbon pool [29–30], while those related to the changes of soil aggregation structure and the physical protection of organic carbon by aggregates with different particle sizes under grazing disturbance is rare, which is not conducive to the elucidation of the mechanism of the effect of grazing disturbance on the soil carbon process of alpine meadows.

In this study, we use the alpine meadow ecosystem as the research background in the Qinghai-Tibet Plateau and use the the typical alpine *Kobresia* swamp meadow as the research object to analyze the influence of different grazing intensities on soil aggregate and the contents of organic carbon in aggregates in the surface soil (0-20cm) of grasslands. The major objectives of this study were the following: (1) to study the changes in allocation of mass and stability of soil aggregate under different grazing intensities: (2) to study the effect of different grazing intensities on the sequestration of organic carbon with different particle sizes.

## Materials and methods

### Description of the study sites

The study area is located in the Gannan Tibetan Autonomous Prefecture of Gansu Province on the eastern Qinghai-Tibet Plateau. It is situated in the Gahai wetland of Luqu County, and its geographical coordinates are 102°05'00''–102°29'45'' E, 33°58'12''–34°30'24'' N. The area is part of the Gahai–Zecha National Nature Reserve. It has a typical continental climate with an altitude of 3430–4300 m, an annual precipitation of 782 mm and an average annual temperature of 1.2°C. The main forage species in this area are *Cyperaceae*, including *Kobresia myosuroides*, *Kobresia tibetica*, *Elymus nutans L.*, *Thalictrum aquilegifolium L. var. sibiricum Regal*, *Carex moorcroftii*, *Blysmus sinocompressus* and *Poa subfastigiata Trin*. The soil types mainly include black meadow soil, boggy soil, and peat soil. The region has a cold climate, humid habitat and abundant rainfall, typical of swamp meadow and alpine meadow areas. In the study area, the grass-growing period is only 120 days because during the grass-growing period, the grasses turn green in early May and tend to become yellow from September to October. In recent years, due to overgrazing and climate warming, the alpine swamp meadow has shown a trend of degrading to secondary bare land.

### Experimental design

Currently, there are some limitations and uncertainties related to defining the disturbance intensity of yak grazing activities at home and abroad [31]. Based on previous research methods, this study fully considered factors such as stock species, livestock grazing behaviors, and time allocation patterns. The summer pastures of three herdsmen families with the same grassland type (Alpine *Kobresia* swamp meadow) and a no grazing area were selected as the study sample near the Gannan Gahai Wetland Protection and Management Station in August 2016. The pasture production and utilization modes of the three herdsmen families were the same (long-term yak grazing), and the grazing period was from June to October every year. A no grazing area was used as a control plot; this area was prohibited from grazing in the core area of the Gahai–Zecha National Nature Reserve for many years (approximately 20 years of grazing prohibition). The four study plots are all located around Gahai Lake. The distances of each plot from Gahai Lake was essentially the same and their altitudes and terrains are also essentially the same (Fig 1). Based on the biomass on the pasture grassland, the theoretical feed

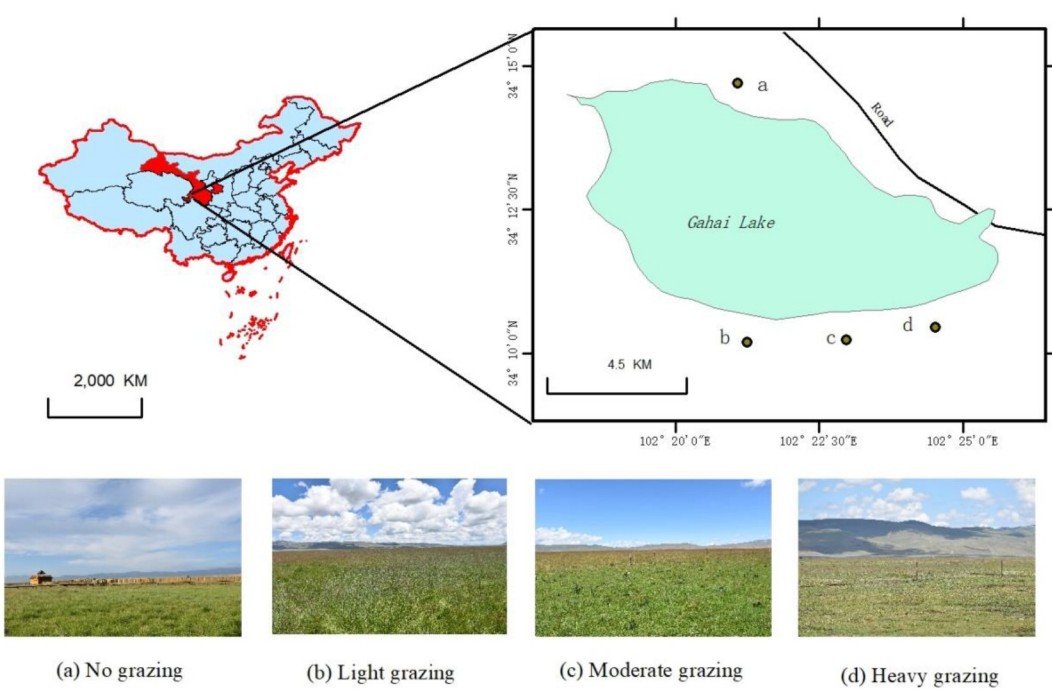

**Fig 1. Overview of the research area.**

intake of yak, grazing time, herd size for many years (the herd sizes of the three herdsmen are essentially stable each year) and the area of fenced grassland, the stocking rate (SR) of yak was determined (Table 1). We set three grazing systems, which are light grazing (LG, SR = 0.8), moderate grazing (MG, SR = 1.8) and heavy grazing (HG, SR = 3.4). No grazing (CK, SR = 0) was used as the control (Table 1). The dominant species of grassland plants were mainly in the genera *Kobresia* and *Cyperus*, and the vegetation distribution of the whole grassland was relatively uniform before 1995. The abovementioned grassland was divided between the three herdsmen and was fenced for grazing after August 1995. Due to the different management strategies of each herdsman, factors such as livestock carrying capacity were different. After more than 20 years of cumulative effects, the surface characteristics and the plant community structure of the grassland in each area were obviously different, resulting in the formation of a natural grazing gradient. See Table 1 for a sample plot overview.

We established three 50 m × 50—m typical sampling sections in every grazing intensity grasslands and randomly establish ten 1 m × 1 m survey samples in each sampling section, with

**Table 1. The site features of Alpine meadow.**

| Grazing intensity | Expermental plot/ hm² | Number of Yaks | Community height/cm | Coverage/% | Stocking rate/ yaks·hm⁻² | Species composition |
|---|---|---|---|---|---|---|
| CK | 25.2 | 0 | 15.15 | 95.34 | 0 | The dominant species is *Kobresia* of *Cyperaceae* |
| LG | 35.3 | 28 | 17.21 | 80.34 | 0.8 | The dominant species is *Gramineae*, and there are many *Kobresia* plants |
| MG | 33.7 | 61 | 8.54 | 58.32 | 1.8 | The dominant species are *Compositae* and *Gramineae*, and poisonous weeds begin to invade in large quantities |
| HD | 20.6 | 70 | 4.11 | 20.11 | 3.4 | Large area of bare surface with a small number of *Rosaceae* and *legume* plants |

a total of 120 samples. The plant species in each area were determined based onmorphology and taxonomy and the height and coverage of each plant in each survey sample were recorded.

## Soil field sampling

Soil sample collection was carried out for each $1 \times 1$-m survey sample. During the sampling, five sampling points were set in each sample according to the "s" shape; then, the litter on the soil surface was removed, and the soil samples from the soil layer extending from 0–20 cm were collected with a soil drill. The soil samples from 5 sampling points were mixed into 1 soil sample (there were 30 mixed soil samples for each grazing sample plot), which was put into the sampling box to avoid crushing and damaging the soil structure. A total of 120 mixed soil samples were taken back to the laboratory for the determination of soil physical and chemical properties and aggregate properties. At the same time, an aluminum box and cutting ring of 100 cm$^3$ were used to collect soil samples at the above sampling points, sample collection was repeated 3 times, and the samples were brought back to the laboratory. Conventional methods were used to measure soil pH, soil bulk density, soil moisture and soil organic matter content [32]. A laser scattering particle analyzer was used to determine the soil particle size distribution (Mastersizer 2000, Malvern Corp, UK). The soil sample was ground by a ball mill after air drying (Mixer Mill MM 200, Retsch GmbH, Germany), and an elemental analyzer (Vario EL III, Elementar, Germany) was used to determine the soil organic carbon content. See Table 2 for the specific measured indexes used under different stocking rates.

## Separation of soil aggregates and soil aggregate stability

The air-dried residual soil sample was sieved through an 8-mm sieve to remove roots and stones. First, we used soil sieves to obtain soil aggregates with a particle size of 3–5 mm. Aggregate stability was measured using the ISO 10930 (2012) method derived from Le Bissonnais [33]. The test was performed on soil subsamples of manually selected 3–5-mm-diameter aggregates (50.00 g each) that were dried at 40°C for 24 h before the test was conducted. After drying, Aggregates were rapidly immersed in 50 ml of deionized water for 10 min. Then the residual aggregates were collected and transferred onto a 0.05-mm sieve previously immersed in ethanol (95%), which was gently hand-moved in a helical movement five times with constant amplitude (4 cm) and frequency (1 s per cycle) each time. The remaining aggregates on the sieve were collected and dried at 40°C for 48 h, and gently dry-sieved using a column of six sieves: 2.00, 1.00, 0.50, 0.20, 0.10 and 0.05 mm. Finally, 7 particle size aggregates were weighed. The treatment was replicated three times.

**Table 2. Changes of soil physical properties under different grazing intensities.**

| Grazing intensity | Soil mechanical composition (%) | | | pH | Soil bulk density (g·cm⁻³) | Soil moisture content (%) | Organic carbon (g·kg⁻¹) | Soil Organic Matter (g·kg⁻¹) |
|---|---|---|---|---|---|---|---|---|
| | Sand particle | Silt particle | Clay particle | | | | | |
| | 2~0.05 mm | 0.05~0.002 mm | < 0.002 mm | | | | | |
| CK | 59.00B | 23.22A | 17.78A | 6.26B | 0.38C | 40.36A | 60.15A | 94.46B |
| LG | 57.05B | 22.74A | 20.21A | 6.45B | 0.51C | 35.13B | 62.36B | 114.27A |
| MG | 65.68A | 18.73B | 15.57B | 7.14A | 0.92A | 8.63C | 37.16C | 69.22C |
| HG | 69.16A | 17.25B | 13.60B | 7.21A | 0.63B | 5.27D | 26.53D | 50.81D |

Different capital letters in the same column indicated significant difference among different grazing intensity at 0.05 level.

The mass (m) of each particle size aggregate was divided by the mass (m) of the whole soil, which is the mass percentage (%) of each particle size aggregate, which is presented as $mp1$ (> 2- mm mass percentage of aggregate), $mp2$ (2–1-mm mass percentage of aggregate), $mp3$ (1–0.5-mm mass percentage of aggregate), $mp4$ (0.5–0.2-mm mass percentage of aggregate), $mp5$ (0.2–0.1-mm mass percentage of aggregate) $mp6$ (0.1–0.05-mm mass percentage of aggregate) and $mp7$ (< 0.05-mm mass percentage of aggregate). Aggregate stability for each sample is expressed as the aggregate mean weight diameter (MWD, mm):

$$mp(i) = \sum \frac{m(i)}{M} \times 100\% \tag{1}$$

Where $mp$(i) refers to the mass percentage of the each particle size aggregate; $m$(i) is the mass of each particle size aggregate; $i$ is the number of each particle aggregate (1–7 refer to aggregates > 2 mm, 2–1 mm, 1–0.5 mm, 0.5–0.2 mm, 0.2–0.1 mm, 0.1–0.05 mm and < 0.05 mm, respectively); and $M$ refers to the quality of the whole soil used for agglomerate classification. In our study, $M$ = 50.00 g.

Currently, there are many methods to determine the stability of soil aggregates, including the mass proportion of water-stable aggregates, aggregate formation index, aggregate failure index, mean weight diameter (MWD) and geometric mean diameter (GMD) [34]. Among them, the MWD is the most important and commonly used index to characterize the stability of aggregates [35], and it is a comprehensive parameter for the particle size distribution of soil aggregates, thus serving as an index to characterize the whole soil structure [36]. The higher the MMD value is, the higher the degree of soil aggregation and the stronger the stability of the aggregates [35]. The calculation method is as follows:

$$MWD = \sum_{i=1}^{n} X_i M_i \tag{2}$$

where X is the mean diameter between two sieves (mm); M is the mass fraction of the aggregates remaining on the sieve (%); I is the number of each particle aggregate, and n is the number of particle-size groups. After the stable 'aggregates' retained on the 2-mm sieve were broken down, the remaining coarse material (gravel > 2-mm) that was retained in the sieve was weighed, and a corrected MWD value without gravel was calculated.

Each MWD value was classified into one of five classes of stability: (i) > 2.0 mm corresponds to very stable material, (ii) 1.3–2.0 mm corresponds to stable material, (iii) 0.8–1.3 mm corresponds to median stability, (iv) 0.4–0.8 mm corresponds to unstable material and (v) < 0.4 mm corresponds to very low stability [33].

## Statistical analysis

One-way analysis of variance (ANOVA) was used to compare the differences in allocation of mass and stability of soil aggregate, as well as aggregate-associated organic carbon of grassland with different grazing intensities (P = 0.05). The LSD method was used for the significance test, and a significance level of P < 0.05 was selected. In order to meet the requirements of the Homogeneity of variance test, we performed a logarithmic transformation of the data. We have used Welch's test to analyze the data when variance uneven. The correlation between different factors was analyzed with the Pearson correlation coefficient method, and the significance level was set as α = 0.05. Using the SPSS 23.0 statistical package(SPSS Inc., Chicago, IL, US)and Microsoft Excel 2010.

## Result

### Characteristics of soil aggregates under different grazing intensities

**Mass and distribution of soil aggregates.** Statistical analysis of the mass of soil aggregates with different particle sizes in the surface layer (0–20 cm) of grassland under different grazing intensities (Table 3). The results showed the following: (1) the primary aggregate size of particles in the grasslands was > 0.2-mm; the sum of macro aggregate masses was between 50.12% and 76.93%, while the mass proportion of micro aggregates (< 0.2-mm) was relatively small. (2) The masses of > 2-mm and 1–2- mm classes of particles in lightly grazed pastures were significantly higher than those of other grazed pastures (P < 0.05), while the mass of particles in the < 0.05-mm class was significantly lower than that in the other grazed pastures (P < 0.05). There were significant differences between different grazed pastures (P < 0.05). (3) With the increase in grazing intensity, the mass of > 2-mm and 2–1-mm classes of particles decreased significantly (P < 0.05), while the mass of the < 0.05-mm class increased significantly (P < 0.05), and that of the other particle size classes did not change significantly. Thus, light grazing is found to be conducive to the formation of macro aggregates in the soil; however, as the grazing intensity continues to increase, the macro aggregates begin to break up and gradually transform into micro aggregates.

**The stability of soil aggregate.** As shown in Table 3, the soil aggregate stability strongly differed in surface soil among the four grazing intensity grasslands. According to the scale defined by Le Bissonnais [33], the aggregate stability was on average (i) moderate (1.08) in the CK and stable (1.33) in the LG grassland, which was significantly higher than that resulting from the other two treatments, indicating that the light grazing of grassland improved soil structure stability and erosion resistance. (ii) With the continuous increase in grazing intensity, the stability index of the soil aggregates changed significantly (P < 0.05), from stable (1.33) in the LG grassland to medium (0.86) in the MG grassland to low (0.64) in the HG grassland, which indicated that MG and HG reduced agglomeration of soil. The stability index of soil aggregates in moderate and severe grazing grasses decreased by 31% and 108% compared with light grazing grasslands, respectively, which leads to the gradual breaking of macro aggregates and finally hinders the formation of good soil structure.

The Pearson correlation analysis method was used to analyze the correlation of MWD with the mass of different particle size aggregates and soil organic carbon content (Table 4). The results showed that MWD was positively correlated with soil organic carbon content (P < 0.01), indicating that the higher the soil organic carbon content is, the greater the MWD value, and the more stable the soil structure. The MWD had the most significant positive correlation with the mass of particles > 2- mm and 2–1-mm (P < 0.01), and had a most negative significant correlation with the mass of particle < 0.05- mm and 0.1–0.05-mm (P < 0.01). This

**Table 3. The allocation of mass of soil aggregates and mean weight diameter index under different grazing intensity(%).**

| Grazing intensity | Composition of soil water-stable aggregates (%) | | | | | | | MWD/mm |
|---|---|---|---|---|---|---|---|---|
| | >2 mm | 2–1mm | 1–0.5mm | 0.5–0.2mm | 0.2–0.1mm | 0.1–0.05mm | <0.05mm | |
| 0 | 42.24B | 10.83B | 5.96B | 3.73C | 6.73B | 8.17B | 22.34C | 1.08B |
| 0.8 | 51.62A | 15.13A | 5.76B | 4.42B | 4.12C | 4.26C | 14.69D | 1.33A |
| 1.8 | 30.62C | 9.63C | 7.27A | 5.20A | 7.69A | 12.71A | 26.88B | 0.86C |
| 3.4 | 22.22D | 6.61D | 7.13A | 5.87A | 8.38A | 12.87A | 36.92A | 0.64D |

Different capital letters in the same column indicated significant difference among different grazing intensity at 0.05 level.

**Table 4. Correlation between parameters of aggregation.**

| Index | Size (mm) | | | | | | | SOC | MWD |
|---|---|---|---|---|---|---|---|---|---|
| | >2 mm | 2–1mm | 1–0.5mm | 0.5–0.2mm | 0.2–0.1mm | 0.1–0.05mm | <0.05mm | | |
| >2 mm | 1 | 0.960** | -0.64 | 0.59 | -0.81* | -0.85* | -0.803* | 0.98** | 0.99** |
| 2–1mm | | 1 | -0.57 | 0.50 | -0.84 | -0.86* | -0.585 | 0.91* | 0.97** |
| 1–0.5mm | | | 1 | 0.35 | -0.55 | 0.53 | -0.263 | -0.64 | -0.63 |
| 0.5–0.2mm | | | | 1 | 0.39 | 0.37 | 0.068 | -0.66 | -0.54 |
| 0.2–0.1mm | | | | | 1 | 0.86 | 0.462 | -0.73 | -0.86 |
| 0.1–0.05mm | | | | | | 1 | 0.682* | -0.79* | -0.91** |
| <0.05mm | | | | | | | 1 | -0.89** | -0.97** |
| SOC | | | | | | | | 1 | 0.96** |
| MWD | | | | | | | | | 1 |

Shown are the mean weight diameter(MWD), Soil organic carbon(SOC).

"*"Indicate a significant level at 0.05,

"**"indicate a significant level at 0.01.

result indicates that the change of these aggregates can reflect the change trend of soil stability in the study area. The content of organic carbon significant affected the turnover between aggregates of particles < 0.1-mm and aggregates of particles > 1-mm in the study area. Their content can be used as an index to measure soil stability.

**The mass of aggregates- associated organic carbon under different grazing intensities.** As shown in Fig 2, the content of organic carbon in the > 2-mm and < 0.05-mm classes of aggregates was the highest in pastures and the organic carbon content varied from 33.22 g·kg$^{-1}$ to 75.62 g·kg$^{-1}$ and 30.92 g·kg$^{-1}$ to 67.69 g·kg$^{-1}$, respectively. The organic carbon content of soil aggregates with particle sizes of > 2-mm, 2–1-mm and < 0.05-mm showed the same changing trend with an increase in grazing, LG > CK > MG > HG, and there were significant

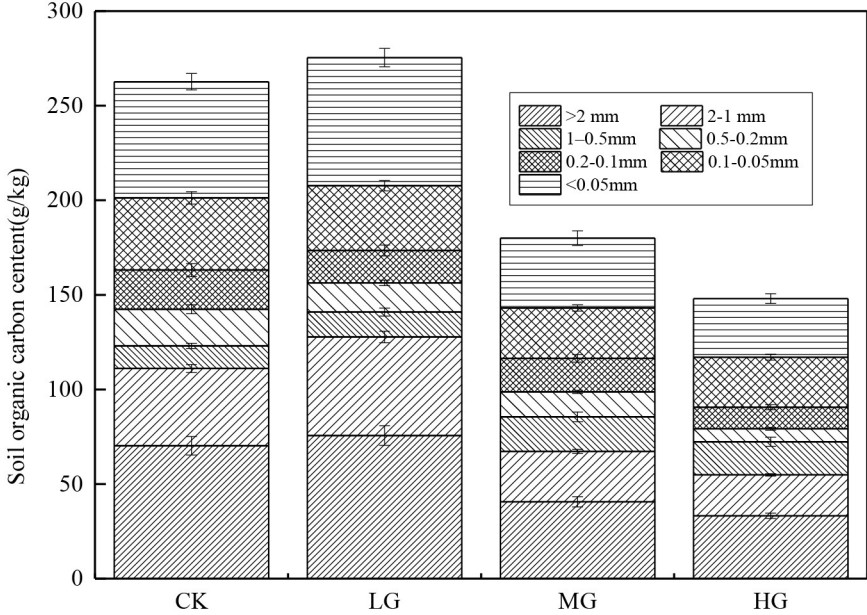

**Fig 2. The allocation of mass of aggregates-associated soil organic carbon of different particle sizes under different grazing gradients.**

differences between different grazing gradients (P <0.05). The change trend in the SOC content in aggregates with particle sizes of 1–0.5-mm was MG > HG > LG > CK, but there was no significant difference between grazing gradients (P > 0.05). The organic carbon content of soil aggregates with particle sizes of 0.5–0.2-mm, 0.2–0.1-mm and 0.1–0.05-mm also exhibited the same change trend with an increase in grazing, shown as MG> HG> LG> CK, but there was no significant difference between grazing gradients (P > 0.05).

## Discussion

### Effect of different grazing intensities on soil aggregates

Soil aggregates are the basic material and functional units of soil and their size and distribution can objectively reflect the soil's ability to supply and store organic carbon and the stability of soil structure during grassland grazing [37]. Marbet et al. [38] noted that aggregates are closely related to organic matter and the increase in soil organic matter content would promote the formation of macro aggregates and the improvement of soil stability. According to the allocation of mass of soil aggregates in different grassland with grazing intensity, the > 0.2-mm particle class of aggregates constituted the majority constituent, and the soil aggregation effect was obvious. This effect may have occurred because 85% of the soil roots of the *Kobresia* meadow in the Gahai wetland were distributed within the 0–20-cm soil layer [39]. The soil has a unique bedding structure in this area, i.e., a sod layer and a humus layer. The cementation and entanglement of plant roots may promote the formation and stability of macro aggregates in the soil in this area, which is consistent with the research conclusions of scholars such as Six [40] and Tisdall [33]. From the change in aggregate content among grazing gradients, the following can be observed: (1) Light grazing significantly increased the content of macro aggregates at particle sizes of > 1-mm, indicating that light grazing promoted the formation of large aggregates and it was conducive to the improvement of soil structure stability. This may be due to the gradually degradation of *Kobresia*, a dominant sedge species, while the niche of *Elymus* increased significantly, which significantly increased the content of soil organic matter and promoted the formation of macro-aggregates with p > 1-mm article sizes. This result may also be due to that the fact that the decomposition speed of plant residues and litter in the alpine environment; the trampling of hoofed animals with a certain intensity level accelerated the decomposition of litter and rotten roots and improved the contact between organic matter and soil, thus promoting the formation of macro aggregates [41]. This is consistent with the viewpoint proposed by Six et al., that "the input of fresh organic residues will promote the formation of large aggregates in soil" [40]. (2) With the continuous increase in grazing intensity, the alpine swamp meadow showed a trend toward gradual transformation from macro aggregate to micro aggregate; moderate and severe grazing activities significantly reduced the proportion of >1-mm-sized particles of macro aggregates in the soil; and the proportion of < 0.05-mm-sized particles in microaggregates trended toward a gradual increase. This was similar to the conclusions reported by Xue [42], who conducted research on alpine meadows in the northeastern Qinghai-Tibet Plateau. The result may be due to the excessive feeding and trampling of pastures by the yak, which leads to a decrease in organic matter input, a gradual increase in soil bulk density, and a reduction in the formation of large aggregates. Finally, macro aggregates gradually change to micro aggregates of < 0.05-mm-sized particles [43]. This result may also be due to the low vegetation coverage and the lack of vegetation protection in the surface soil. Frequent dry-wet cycles and freeze-thaw effects change the structure and composition of soil aggregates, resulting in the fragmentation of macro aggregates into micro aggregates [44].

The MWD value is a common indicator that reflects the size distribution of soil aggregates. The larger the value is, the higher the soil agglomeration, and the greater the stability [35]. In this study, we found that grazing prohibition and lightly grazing grasslands have higher MWD values, indicating good soil structure stability and strong anti-erosion ability. The main reason may be because there is more surface litter reserve, which increased the content of organic cementing substance in the soil. In addition, the soil bulk density of the grassland was small, the soil structure was loose, and the aeration was good, which was conducive to the formation of macro aggregates, thereby improving the stability of the soil structure. From light grazing to moderate grazing, the MWD value decreased significantly, and soil stability became worse, which indicated that light grazing was an important turning point in the decline in soil aggregate stability, and the continuous increase in grazing pressure would hinder the formation of good soil structure. This is mainly because overgrazing reduced macro aggregation of soil, causes the disintegration of soil aggregates, and ultimately leads to the decline in aggregate stability [43]. The study also found that the stability of soil aggregates in light grazing grasslands was also found to be higher than that in nongrazing grasslands, in contrast to the results from the study on alpine meadows in the Qinghai Tibet Plateau by Xue et al. [42], indicating that light grazing was beneficial to the formation of good soil structure. Pearson correlation analysis found that MWD was significantly positively correlated with soil organic carbon content, indicating that soil organic carbon content affects the body stability of soil aggregates. According to the Pearson correlation analysis, MWD and aggregates with particle sizes > 1-mm and < 0.01-mm was significant, which indicated that the content of these aggregates in the soil had a greater impact on the stability of soil aggregates, while other aggregates had weak impact.

## Effect of different grazing intensities on aggregates-associated organic carbon

Elliott [45], Jastrow [46] and other studies found that macro aggregates contain more organic carbon than micro aggregates, which is inconsistent with our research results. In this study, the content of soil organic carbon in the aggregates consisting of > 2-mm—and < 0.05-mm—sized particles was higher in every grazing grassland, which was similar to the distribution characteristics of soil aggregate organic carbon in the degraded alpine meadow of the northern Tibet Plateau studied by Yu [47]. The content of organic carbon in the aggregates > 2-mm-sized particles was high, which may be related to the decomposition of fresh organic carbon residues. The organic carbon formed by decomposition will first gather in the macro aggregate and the organic substances in the decomposed state in the macro aggregates greatly increased the concentration of organic carbon. Therefore, the content of organic carbon in aggregates of particle size > 2-mm was relatively high [40]. The content of organic carbon in the aggregates of < 0.05-mm-sized particles was also high. However, it is possible that macroaggregates are decomposed by microorganisms in the cycle of soil nutrients, causing the breakdown of macroaggregates and the release of microaggregates, which results in an enrichment of soil organic carbon in < 0.05-mm-sized particle aggregates [48], similar to the "macroaggregate turnover conceptual model" proposed by Six et al. [40]. However, it is possible that the organic and inorganic colloids in the microaggregates can closely combine fixed carbon, which is an inert organic carbon component with high humification that is not easily decomposed and released by microorganisms. In addition, the cold and humid environment of the Qinghai-Tibet Plateau is not conducive to the decomposition of microorganisms, which leads to the accumulation of organic carbon in the aggregates [21]. The study also showed that with the increase in grazing intensity, the organic carbon content in aggregates with > 2-mm-, 1-2-mm- and

< 0.05-mm-sized particles reached the maximum in light grazing grasslands and then gradually decreased. This was related to the high content of soil organic matter, the good stability of soil aggregate and the strong activity of microorganisms in the light grazing grasslands. In moderately and heavily grazed grasslands, the input of fresh organic matter is less, which may disrupt the balance of input and output of original soil organic carbon. The microorganisms in the soil rely more heavily on the original organic matter in the soil as energy, thus accelerating the decomposition of soil organic cementation materials, leading to the gradual disintegration of aggregates and the continuous mineralization of organic carbon by microorganisms, thereby reducing the size of each particle cluster. thus reducing the content of organic carbon in aggregates of different particle sizes.

## Conclusion

The aggregates of > 0.2-mm-sized particles occupied the majority of the soil surface layer (0–20 cm) of the swamp meadow in Gannan Gahai wetland and the light grazing activities increased the mass of macro aggregates composed of particles with diameters > 2-mm and 1–2-mm and increased the stability of soil aggregates. In contrast, moderate and severe grazing activities led to the transformation of macro aggregates into micro aggregates. Aggregates of particle size > 2-mm and < 0.05-mm had a strong fixed effect on organic carbon. The light grazing promoted considerably the content of organic carbon in the aggregates of the particle size classes described above, while moderate and heavy grazing reduced the content of organic carbon in aggregates, leading to the breakage of macro aggregates and the decrease of soil structural stability, thereby accelerating the decomposition of organic carbon. This may reduce the carbon storage in alpine marsh meadow soil. Light grazing is a reasonable yak grazing system in the study area.

## Prospective

Currently, there are some limitations and uncertainties in the definition of the interference intensity of yak grazing activities. In the future, we should set up multiple grazing interference intensities in the study area and carry out research on soil structure and soil carbon sequestration. Determination of the most reasonable grazing intensity of grassland in the study area can provide a valuable reference for the coordinated development of animal husbandry and swamp meadow ecosystems in the eastern part of the Qinghai -Tibet Plateau.

There are also some limitations to this study that should be noted. The experimental design of this study did not take into account the weight, age or daily grazing duration of yaks, which may have impacted the experimental results.

## Supporting information

**S1 File. Data in the manuscript.**
(XLSX)

## Author Contributions

**Conceptualization:** Jiwei Wang, Chengzhang Zhao.

**Methodology:** Chengzhang Zhao, Lianchun Zhao.

**Writing – original draft:** Jiwei Wang.

**Writing – review & editing:** Jun Wen, Qun Li.

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
