## [Decision Letter · Decision Letter 0]

17 Mar 2020

PONE-D-20-04653

Effects of Grazing Intensity on Soil Water-stable Aggregates and Their Stability and Organic Carbon Fractions in an Alpine Swamp Meadow in the East Qinghai-Tibet Plateau

PLOS ONE

Dear Mr Wang,

Thank you for submitting your manuscript to PLOS ONE. After careful consideration, we feel that it has merit but does not fully meet PLOS ONE’s publication criteria as it currently stands. Therefore, we invite you to submit a revised version of the manuscript that addresses the points raised during the review process.

Should you decide to revise the paper, please be aware that a thorough revision of the whole manuscript is necessary, including English style.

We would appreciate receiving your revised manuscript by May 01 2020 11:59PM. To enhance the reproducibility of your results, we recommend that if applicable you deposit your laboratory protocols in protocols.io, where a protocol can be assigned its own identifier (DOI) such that it can be cited independently in the future. For instructions see: http://journals.plos.org/plosone/s/submission-guidelines#loc-laboratory-protocols

We look forward to receiving your revised manuscript.

Kind regards,

Remigio Paradelo Núñez

Academic Editor

PLOS ONE

Journal Requirements:

3.  We note that [Figure 1] in your submission contains a  [map/satellite] image which may be copyrighted. All PLOS content is published under the Creative Commons Attribution License (CC BY 4.0), which means that the manuscript, images, and Supporting Information files will be freely available online, and any third party is permitted to access, download, copy, distribute, and use these materials in any way, even commercially, with proper attribution. For these reasons, we cannot publish previously copyrighted maps or satellite images created using proprietary data, such as Google software (Google Maps, Street View, and Earth). For more information, see our copyright guidelines: http://journals.plos.org/plosone/s/licenses-and-copyright.

1.     You may seek permission from the original copyright holder of Figure [1] to publish the content specifically under the CC BY 4.0 license.  

Additional Editor Comments (if provided):

The manuscript reports the effect of grazing intensity on SOC distribution among water-stable aggregates in grassland soils from the Tibet. As highlighted by the reviewers, English style should be thoroughly revised. In addition to this, there are many improvements in scientific style that must be made:

- the discussion is too long and should be more focused;

- there is almost no information about soil properties, such as pH, texture, classification… Differences in soil nature among points could influence the results

- tables and figures need to be self-explicative; also, it is not clear in Figures 2-4 what is plotted as the SOC-LFOC-POC sections of the bars

- the objective of the stepwise regression analysis performed is not clear

Reviewers' comments:

Reviewer's Responses to Questions

**Comments to the Author**

1. Is the manuscript technically sound, and do the data support the conclusions?

Reviewer #1: Yes

Reviewer #2: Yes

2. Has the statistical analysis been performed appropriately and rigorously? 

Reviewer #1: No

Reviewer #2: N/A

3. Have the authors made all data underlying the findings in their manuscript fully available?

Reviewer #1: Yes

Reviewer #2: Yes

4. Is the manuscript presented in an intelligible fashion and written in standard English?

Reviewer #1: Yes

Reviewer #2: No

5. Review Comments to the Author

Reviewer #1: The eastern part of the Qinghai-Tibet Plateau is an ecologically fragile area, and it is also a main carrying area of China's alpine livestock industry. How to coordinate the relationship between the development of animal husbandry and ecological environment protection is the main problem facing sustainable development in this region. This article uses experimental research methods to study the changes in soil aggregates and organic carbon under different grazing intensities, and provides a valuable reference for the rational planning of grazing intensities in the alpine pastoral area of the eastern Qinghai-Tibet Plateau. At the same time, the experimental data and results in the paper are of reference value for similar research. However, the current MS focuses on too many parts which are not well connected, and the emphasis is not outstanding. Here are the detailed comments:

1. Title is too long and research is not focused enough. This study involves grazing intensity, biomass, soil water-stable aggregates, SOC, LFOC, POC, MWD, etc., but these aspects have not been well connected, making the focus of the paper ambiguous. I suggest shortening the title to focus on one scientific issue.

2. Above-ground and underground biomass were measured in the study, how does this part of the study support for the conclusions of the paper? If biomass study does not contribute much to the core content, it is recommended to delete it.

3. The conclusion section does not sufficiently summarize the content of the results and discussion. It is suggested to answer the three questions raised in the introduction according to the content of the research.

4. The keywords section should also include some key words not in the title.

5. The remote sensing image in Figure 1 should also include a scale bar. In the map of China in Figure 1, should there be a nine-segment line in the South China Sea?

6. Many important results in the paper are based on the statistics of the samples, such as Figure 2, 3, 4. The screening of outliers in sample data is very important for the robustness of the results. Similarly, when studying the relationship between MWD and SOC using statistical methods, it is also necessary to identify the possible error samples.

7. It is recommended to combine the three figures of Figure 2, 3, 4 into one large picture, which is also convenient for readers to compare.

8. The abbreviation mentioned for the first time in the body of MS (other than the abstract) should explain its meaning, such as SOC, LFOC, POC, etc.

9. How is SOC classified? LFOC should correspond to HFOC, POC should correspond to WSOC.

10. In Table 2, why there are two capital letters behind the statistics of above-ground biomass with grazing intensity of LG?

11. The formulas should be followed by serial numbers in the MS.

12. At the end of the MS, the future research trends should be prospected, and suggestions should be made for the coordinated development of animal husbandry and wetland swamp meadow ecosystems in the eastern Qinghai-Tibet Plateau.

Reviewer #2: General comments:

The manuscript provides very informative results on grazing Intensity on soil water-stable aggregates in in an Alpine Swamp Meadow in the East Qinghai-Tibet Plateau. However, the main weakness is the English style, which has to be changed in many places.

Detailed Comments：

Line 11-12: it is hard to understand

Line 15：Something missing/unclear, NWD.

Line 15-16: do you mean SOC and its fractions in bulk soil or aggregate?

Line 16-17: I do not find mechanism study about soil aggregate formation in this study; you want to state that the percentage of larger aggregates under no grazing plots was more than

……

As mentioned above, I tried to give many remark as possible; Editing by a native speaker would be highly recommended.

So, I encourage the authors to improve the language and then to resubmit their work.

6. PLOS authors have the option to publish the peer review history of their article (what does this mean?). If published, this will include your full peer review and any attached files.

Reviewer #1: No

Reviewer #2: No

---

## [Author Response · Author response to Decision Letter 0]

4 May 2020

Dear Editors and Reviewers: 

Thank you for your letter and for the reviewers’ comments concerning our manuscript entitled “Effects of Grazing Intensity on Soil Water-stable Aggregates and Their Stability and Organic Carbon Fractions in an Alpine Swamp Meadow in the East Qinghai-Tibet Plateau” (ID: PONE-D-20-04653). We sincerely thank the editor and all reviewers for their valuable feedback that we have used to improve the quality of our manuscript. The reviewer comments are laid out below in italicized font and specific concerns have been numbered. Our response is given in normal font and changed to the manuscript are given in blue highlighting text. The main corrections in the paper and the responds to the reviewer’s comments are as flowing: 

Response to Journal Requirements

1. Comments: Please ensure that your manuscript meets PLOS ONE's style requirements, including those for file naming.

Response: The manuscript has been modified with reference to the magazine model.

2. Comments: In your Methods section, please provide additional information regarding the permits you obtained for the work. Please ensure you have included the full name of the authority that approved the field site access and, if no permits were required, a brief statement explaining why.

Response：We carried out this work in the field observation station of Gansu Agricultural University, a long-term cooperative unit, so we did not need permission to enter the research area.

3. Comments: We note that [Figure 1] in your submission contains a [map/satellite] image which may be copyrighted.

Response: We have redrawed Figure 1. and changed the remote sensing images to vectorgraph.

Figure 1. Overview of the research area

Response to Additional Editor Comments (if provided)

1. Comments: the discussion is too long and should be more focused

We have referred to the recommendations of the first reviewer, and then refocused the focus of research of the manuscript, that only focus on the scientific issue of “Effect of razing on the aggregates and aggregate-associated organic carbon”. At the same time, we deleted the content of light fraction organic carbon (LFOC) and particulate organic carbon (POC) in the manuscript. So on the basis of the original manuscript, we focused on the content in the heading of "Discussion" and rewritten the "Discussion". See lines 261-348 of "Manuscript" for details

2. Comments: there is almost no information about soil properties, such as pH, texture, classification… Differences in soil nature among points could influence the results

Response: We supplemented the soil properties data of four study plots, including pH, soil mechanical composition, soil organic carbon and soil organic matter. Among them, pH and soil organic carbon indexes have been measured in October 2016. Since the collected soil samples are still kept in the laboratory, we have recently tested and analyzed the soil samples to obtain the data of “soil mechanical composition”.

Table 2 Changes of soil physical and chemical properties under different grazing intensities

Table 2. Changes of soil physical properties under different grazing intensities

Grazing intensity Soil mechanical composition (%) pH Soil bulk density

( g•cm-3) Soil moisture content (%) Organic carbon ( g•kg-1) Soil Organic Matter

( g•kg-1)

 Sand particle Silt particle Clay particle 

 2～0.05 mm 0.05～0.002 mm < 0.002 mm 

CK 59.16B 23.16A 17.68A 6.26B 0.38C 40.36A 60.15A 94.46B

LG 57.25B 22.62A 20.13A 6.45B 0.51C 35.13B 62.36B 114.27A

MG 65.63A 18.84B 15.53B 7.14A 0.92A 8.63C 37.16C 69.22C

HG 69.11A 17.28B 13.61B 7.21A 0.63B 5.27D 26.53D 50.81D

Different capital letters in the same column indicated significant difference among different grazing intensity at 0.05 level.

3. Comments: tables and figures need to be self-explicative; also, it is not clear in Figures 2-4 what is plotted as the SOC-LFOC-POC sections of the bars

Response: We deleted the abbreviation "SOC" of soil organic carbon in Figure 2. 

Figure 2. The allocation of mass of aggregates-associated soil organic carbon of different particle sizes under different grazing gradients

4. Comments: the objective of the stepwise regression analysis performed is not clear

Response: With reference to the revision suggestions proposed by the editor on the manuscript, we think that it was not reasonable to analyze the correlation among MWD and organic carbon and organic carbon components in different particle sizes of aggregate by stepwise regression analysis method, because there is no direct correlation between them, so we deleted this part. In our study, soil organic carbon and the mass of aggregates with different particle sizes may be the main factors that cause the stability change of aggregates, so we used Pearson correlation method to study the relationship between these two factors. See lines 236-246 of "Manuscript" for details

Response to Reviewer #1

1. Comments: Title is too long and research is not focused enough. This study involves grazing intensity, biomass, soil water-stable aggregates, SOC, LFOC, POC, MWD, etc., but these aspects have not been well connected, making the focus of the paper ambiguous. I suggest shortening the title to focus on one scientific issue.

Response: In response to the questions raised by the reviewers, after careful consideration, we redefined the focus of the manuscript and deleted the indicators of light fraction organic carbon (LFOC) and particulate organic carbon (POC). We have only focused on the scientific issue of “Effect of grazing on the mass of aggregates and organic carbon content in aggregates” We modified the title to " Effects of Grazing on The Allocation of Mass of Soil Aggregates and Aggregate-associated Organic Carbon in an Alpine Meadow ". See " Revised manuscript with Track Changes " for the corresponding modification in manuscript.

2. Comments: Above-ground and underground biomass were measured in the study, how does this part of the study support for the conclusions of the paper? If biomass study does not contribute much to the core content, it is recommended to delete it.

 Response: we have deleted the data of Above-ground and underground biomass in manuscript.

3. Comments: The conclusion section does not sufficiently summarize the content of the results and discussion. It is suggested to answer the three questions raised in the introduction according to the content of the research.

 Response: Based on the comments made by the reviewers, we have rewritten the “Conclusion”.

Conclusion

The aggregates of > 0.2-mm-sized particles occupied the majority of the soil surface layer (0-20 cm) of the swamp meadow in Gannan Gahai wetland and the light grazing activities increased the mass of macro aggregates composed of particles with diameters > 2 mm and 1-2 mm and increased the stability of soil aggregates. In contrast, moderate and severe grazing activities led to the transformation of macro aggregates into micro aggregates. Aggregates of particle size > 2mm and <0.05 mm had a strong fixed effect on organic carbon. The light grazing promoted considerably the content of organic carbon in the aggregates of the particle size classes described above, while moderate and heavy grazing reduced the content of organic carbon in aggregates, leading to the breakage of macro aggregates and the decrease of soil structural stability, thereby accelerating the decomposition of organic carbon. This may reduce the carbon storage in alpine marsh meadow soil. Light grazing is a reasonable yak grazing system in the study area. 

4. Comments: The keywords section should also include some key words not in the title.

Response: keywords: Qinghai-Tibet Plateau; grazing; soil aggregate; soil organic carbon; particle size

5. Comments: The remote sensing image in Figure 1 should also include a scale bar. In the map of China in Figure 1, should there be a nine-segment line in the South China Sea?

Response: We have redrawed Figure 1. and add a nine-segment line in the Figure 1.

Figure 1. Overview of the research area

6. Comments: Many important results in the paper are based on the statistics of the samples, such as Figure 2, 3, 4. The screening of outliers in sample data is very important for the robustness of the results. Similarly, when studying the relationship between MWD and SOC using statistical methods, it is also necessary to identify the possible error samples.

Response: (1) No error in field sampling. It was sunny in the study area, where has been sunny for one week before sampling. The sampling personnel were all in the same group and the sampling method was the same. So there is no sampling error. (2) There is no error in data analysis. We use the Box-plot of SPSS software to judge whether there was any abnormal value in the data. The results show that there are no outliers in our data. If there are outliers, we will not remove them. We will analyze the reason for the outliers in detail. All the data used in our statistical methods obey the normal distribution. 

7. Comments: It is recommended to combine the three figures of Figure 2, 3, 4 into one large picture, which is also convenient for readers to compare.

Response: Because we focused on the research priorities of the manuscript, all relevant research contents of light fraction organic carbon (LFOC) and particulate organic carbon (POC) were deleted, including figure 3 and Figure 4.

8. Comments: The abbreviation mentioned for the first time in the body of MS (other than the abstract) should explain its meaning, such as SOC, LFOC, POC, etc.

Response: We have revised the manuscript in accordance with the comments of the reviewers. 

9. Comments: How is SOC classified? LFOC should correspond to HFOC, POC should correspond to WSOC.

Response: Your point of view is very correct, LFOC should correspond to HFOC and POC should correspond to WSOC. However, because there are many research contents in the manuscript, the focus is not clear, so we deleted the relevant research content of LFOC and POC in the manuscript.

10. Comments: In Table 2, why there are two capital letters behind the statistics of above-ground biomass with grazing intensity of LG?

Response: I'm so sorry for typos caused by our carelessness. We have been revised them in the manuscript.

11. Comments: The formulas should be followed by serial numbers in the MS.

Response: According to the modification, we have added the serial numbers of the formula in the manuscript.

 (1)

 (2)

12. Comments: At the end of the MS, the future research trends should be prospected, and suggestions should be made for the coordinated development of animal husbandry and wetland swamp meadow ecosystems in the eastern Qinghai-Tibet Plateau.

Response: Thanks to the comments made by the reviewers, we have look forward to the future research priority of the coordinated development of animal husbandry and wetland swamp meadow ecosystems in the eastern Qinghai-Tibet Plateau.

Prospective

At present, there are some limitations and uncertainties in the definition of the disturbance intensity of yak grazing activities. In the future, we should set up multiple grazing disturbance intensities in the study area and carry out the research on soil structure and soil carbon sequestration benefits to explore the most reasonable grazing intensity of grassland in the study area, providing valuable reference for the coordinated development of animal husbandry and swamp meadow ecosystem in the eastern part of the Qinghai-Tibet Plateau. There are also some limitations to this study. The experimental design of this study did not take into account the weight, age and daily grazing duration of yaks, which may have had an impact on the experimental results.

Response to Reviewer #2

1. Comments: Line 11-12: it is hard to understand

Response: We rewrote the abstract, and the professionals retouched the language。

2. Comments: Line 15：Something missing/unclear, NWD

Response: We are very sorry for our carelessness, NWD is a typo, NWD was corrected as MWD (mean weight diameter)

3. Comments: Line 15-16: do you mean SOC and its fractions in bulk soil or aggregate?

Response: SOC and its fractions in aggregate

4. Comments: Line 16-17: I do not find mechanism study about soil aggregate formation in this study

Response: Thank you for your suggestion. We removed the imprecise statement in the abstract.

5. Comments: As mentioned above, I tried to give many remark as possible; Editing by a native speaker would be highly recommended.

Response: Our revised manuscript was edited by a native speaker.

We tried our best to improve the manuscript and made some changes in the manuscript. Such as the "Introduction", "Result" and "Discussion" in the original manuscript are too long. So we focus on them, and we were compressed and rewrote the contents of the "Introduction", "Result" and "Discussion". These changes will not influence the content and framework of the paper. And here we did not list the changes but marked in blue in revised paper.

We appreciate for Editors/Reviewers’ warm work earnestly, and hope that the correction will meet with approval. It there are any other modifications we could make, we would like very much to modify them and we really appreciated your help. Thank you very much for your help.

---

## [Decision Letter · Decision Letter 1]

28 May 2020

Effects of Grazing on The Allocation of Mass of soil Aggregates and Aggregate-associated Organic Carbon in an Alpine Meadow

PONE-D-20-04653R1

Dear Dr. Wang,

We are pleased to inform you that your manuscript has been judged scientifically suitable for publication and will be formally accepted for publication once it complies with all outstanding technical requirements.

With kind regards,

Remigio Paradelo Núñez

Academic Editor

PLOS ONE

Additional Editor Comments (optional):

Reviewers' comments:

Reviewer's Responses to Questions

**Comments to the Author**

1. If the authors have adequately addressed your comments raised in a previous round of review and you feel that this manuscript is now acceptable for publication, you may indicate that here to bypass the “Comments to the Author” section, enter your conflict of interest statement in the “Confidential to Editor” section, and submit your "Accept" recommendation.

Reviewer #1: All comments have been addressed

2. Is the manuscript technically sound, and do the data support the conclusions?

Reviewer #1: Yes

3. Has the statistical analysis been performed appropriately and rigorously? 

Reviewer #1: Yes

4. Have the authors made all data underlying the findings in their manuscript fully available?

Reviewer #1: Yes

5. Is the manuscript presented in an intelligible fashion and written in standard English?

Reviewer #1: Yes

6. Review Comments to the Author

Reviewer #1: (No Response)

7. PLOS authors have the option to publish the peer review history of their article (what does this mean?). If published, this will include your full peer review and any attached files.

Reviewer #1: No

---

## [Editor Report · Acceptance letter]

1 Jun 2020

PONE-D-20-04653R1 

Effects of Grazing on The Allocation of Mass of soil Aggregates and Aggregate-associated Organic Carbon in an Alpine Meadow 

Dear Dr. Wang:

I am pleased to inform you that your manuscript has been deemed suitable for publication in PLOS ONE. Congratulations! Your manuscript is now with our production department. 

With kind regards,

on behalf of

Dr. Remigio Paradelo Núñez 

Academic Editor

PLOS ONE